# Genome-Wide Identification and Expression Characterization of the *D27* Gene Family of *Capsicum annuum* L.

**DOI:** 10.3390/plants13152070

**Published:** 2024-07-26

**Authors:** Di Wu, Wenting Fu, Nanyi Wang, Yong Ye, Jianwen He, Kangyun Wu

**Affiliations:** 1Research Institute of Pepper, Guizhou Academy of Agricultural Science, Guiyang 550025, China; 876865710@aliyun.com (D.W.); fu20210802@outlook.com (W.F.); wangnanyi0202@163.com (N.W.); yeyong0855@126.com (Y.Y.); 2Key Laboratory of Crop Genetic Resources and Germplasm Innovation in Karst Mountain Areas, Ministry of Agriculture and Rural Affairs, Guiyang 550025, China

**Keywords:** *Capsicum annuum*, *D27* gene family, strigolactones, protein–protein interaction, whole-genome identification

## Abstract

As a crucial member of the gene family involved in the biosynthesis of strigolactones, *D27* plays an important regulatory role in plant branching and root development, which is essential for field management and yield increase in peppers (*Capsicum annuum* L.). To comprehensively understand the characteristics of the pepper *D27* gene family, we identified three *CaD27* genes. By analyzing their physicochemical properties, phylogenetic relationships, gene structures, promoters, and expression patterns in different tissues, the characteristics of the *CaD27* gene family were revealed. The research results showed that these three *CaD27* genes are located in three different chromosomes. Evolutionary analysis divided the members of CaD27 into three groups, and gene collinearity analysis did not find any duplicates, indicating the diversity and non-redundancy of the *CaD27* gene family members. In addition, we identified and classified *cis*-elements in the promoter regions of *CaD27* genes, with a relatively high proportion related to light and plant hormone responses. Expression pattern analysis showed that *CaD27.1* is expressed in leaves, while *CaD27.2* is expressed in roots, indicating tissue specificity. Furthermore, protein interaction predictions revealed an interaction between D27.2 and CCD7. This study provided important insights into the function and regulatory mechanisms of the *CaD27* gene family and the role of strigolactones in plant growth and development.

## 1. Introduction

Plant architecture is crucial for the growth and development of plants. The formation of plant architecture is mainly controlled by various factors, including genetics, endogenous hormone levels, nutrient status, temperature and humidity, and photoperiod [1]. In addition, both biotic and abiotic stresses impact the entire life cycle of plants, leading to a decrease in crop yield and quality [2]. Plants respond to unfavorable conditions by regulating their hormone levels, with strigolactones (SLs) playing an important role in the regulation of plant branching, growth, development, and stress responses [3,4]. Studies have shown that strigolactones can inhibit plant branching, shape root morphology, promote leaf senescence, and regulate plant secondary growth [5,6]. Furthermore, they also participate in plant stress responses, including drought tolerance [7], cold tolerance [8], salt tolerance [9], and stress resistance [10,11].

Strigolactones are terpenoid compounds produced in the roots and belong to the carotenoid biosynthesis pathway. Their synthesis involves the participation of at least five enzymes: cis/trans-carotene isomerase DWARF27 (D27), two carotenoid cleavage dioxygenases (CCD7 and CCD8), a cytochrome P450 monooxygenase MORE AXILLARY GROWTH 1 (MAX1), and an oxidoreductase enzyme LATERAL BRANCHING OXIDOREDUCTASE (LBO) [12,13]. As a newly discovered plant hormone, strigolactones have attracted significant attention. They were initially found to induce the germination of the seeds of root parasitic plants [14]. Subsequently, it has been shown that they can also promote fungal branching [15]. Mutants of SLs synthesis genes in peas and rice exhibit increased tillering due to the lack of SLs, while the addition of exogenous SLs analog GR24 can restore this phenotype [3,16]. Furthermore, research has found that GR24 can promote the formation of rhizobium nodules in alfalfa and peas [17,18], as well as promote lateral root formation and increase root hair length in *Arabidopsis thaliana* [19]. SLs can also inhibit adventitious root formation [20,21] and participate in the regulation of leaf senescence and other processes in plants [22]. Additionally, studies have found that SLs synergistically regulate plant growth and development with other hormones.

Recent reports have indicated that strigolactones (SLs) cooperate with abscisic acid (ABA), auxin, cytokinin, and gibberellin to regulate plant growth, development, and stress responses [8,11]. Research has found that ABA8′-hydroxylase significantly promotes ABA accumulation while reducing SLs’ levels in the roots [23]. Under drought or heat stress, exogenous application of GR24 can lower ABA levels in plants [24,25]. SLs and ABA jointly regulate stomatal closure in response to water stress [7,26] and participate in the regulation of the flavonoid biosynthesis pathway to enhance plant drought tolerance [27]. Moreover, SLs and auxin synergistically regulate plant branching. SLs influence the biosynthesis and transport of auxin, thereby regulating shoot growth and development, while auxin actively regulates SLs biosynthesis [28,29,30]. SLs affect shoot branching by modulating auxin transport [31]. Low concentrations of GR24 promote shoot branching, but high concentrations reduce branching [32]. The interaction between SLs and cytokinins (CK) exhibits tissue specificity, independently regulating adventitious roots [20], antagonistically regulating axillary bud activation and branching, and synergistically participating in lateral root development [33]. The molecular mechanisms of SLs and gibberellin (GA) signal transduction are similar, with both relying on α/β-hydrolases and E3 ubiquitin ligases-mediated protein degradation [34]. In *Arabidopsis thaliana*, SLs can inhibit ABA synthesis and accumulation and promote GA synthesis, thereby reducing the ABA/GA ratio under heat stress [25]. Overall, SLs have been proven to be closely associated with plant growth, development, and stress responses.

Pepper (*Capsicum annuum* L.) is an important vegetable and spice, widely cultivated worldwide. It is rich in vitamins, carotenoids, and capsaicin compounds [35]. The growth, development, and stress responses of pepper significantly affect its yield and quality. Strigolactones (SLs) are key plant hormones involved in pepper growth and stress tolerance [5,10]. The *D27* gene encodes a β-carotene isomerase, which participates in the initial steps of strigolactone biosynthesis. The D27 enzyme isomerizes β-carotene to 9-cis-β-carotene, a precursor for strigolactone synthesis [12,13]. Although the *D27* gene has been identified in many plants, this has not been identified in pepper. Therefore, we conducted a whole-genome identification of *D27*-related genes in the pepper genome. We analyzed the gene structure, chromosome distribution, promoter *cis*-elements, and phylogenetic relationships of *D27*-related genes. Additionally, we examined the expression patterns of these genes in different organs and verified their protein–protein interactions through yeast two-hybrid assays. The research results provide valuable information for studying pepper growth, development, and the functions of *D27* genes.

## 2. Results

### 2.1. Identification and Physicochemical Property Analysis of the CaD27 Gene Family

Through HMMER and BLAST searches, we identified three *D27* genes in the “Zunla” pepper genome, located on chromosomes 1, 3, and 6, and named *CaD27.1*, *CaD27.2*, and *CaD27.3*, respectively (Figure 1). The number of amino acids in *CaD27* ranges from 227 to 261, with molecular weight of 26.0–28.8 kDa. The isoelectric points range from 6.61 to 8.46. The protein instability index is greater than 40, indicating that they are unstable proteins. Subcellular localization predictions show that all three members of the *D27* gene family are located in the chloroplast (Appendix A).

### 2.2. Collinearity Analysis of the CaD27 Gene Family

In plant evolution, replication events play an important role in the amplification of gene family members. Firstly, intraspecific collinearity analysis suggests that the “Zunla” pepper does not possess any *CaD27* collinearity genes, indicating that the gene sequences among the members of the *CaD27* gene family are not repeated and exhibit diversity (Figure 2A).

The collinearity analysis of the *CaD27* genes between the “Zunla” pepper and the “Ca59” and “CM334” pepper varieties reveals three and two pairs of colinear members, respectively, indicating that *D27* is relatively conserved and homologous among the three pepper varieties. On the other hand, the collinearity analysis of the “Zunla” pepper with *Arabidopsis thaliana* and other Solanaceae plants shows three, one, two, and one pairs of colinear members with *Solanum lycopersicum*, *Arabidopsis thaliana*, *Solanum tuberosum*, and *Nicotiana tabacum*, respectively (Figure 2B). This suggests that the D27 protein of “Zunla” pepper is relatively conserved and most similar to *Solanum lycopersicum*, followed by *Solanum tuberosum*.

### 2.3. D27 Gene Family Phylogenetic Analysis

To evaluate the evolutionary relationship between *Capsicum annuum* (Zunla) *D27* members and the corresponding proteins in other plants, a phylogenetic tree was constructed using their protein sequences and those of other plants (Appendix A). All D27 proteins can be classified into three main groups: Group A, Group B, and Group C (Figure 3). The three CaD27 members belong to different groups, indicating diversity among them and suggesting differences in their functions or structures. Among them, CaD27.2 is located in Group A and shares a closer evolutionary relationship with *Solanaceae*, consistent with the results of the collinearity analysis, suggesting that the CaD27.2 member has functional similarity to the D27 protein in *Solanaceae*. CaD27.1 is in Group B and is closely related to *Lycium barbarum* (*LbD27*), while CaD27.3 is in Group C and is most closely related to *Solanum pennellii* (*SpD27*). *Lycium barbarum* (*LbD27*), *Solanum pennellii* (*SpD27*), and *Capsicum annuum* all belong to the Solanaceae family, indicating a higher degree of similarity among them.

### 2.4. Analysis of Codon Preference in the CaD27 Gene Family

Analysis of codon preference in the *CaD27* gene family reveals that the GC content ranges from 0.367 to 0.456, with an average value of 0.425. The third base (GC3s) shows a preference for G or C, with a frequency ranging from 0.313 to 0.435 and an average value of 0.382. Regarding the third base of codons, the average frequencies of A, T, G, and C (A3s, T3s, C3s, G3s) are 0.37, 0.42, 0.21, and 0.30, respectively. The effective number of codons (ENc), which reflects the degree of codon preference in a gene, ranges from 21 to 60. The closer the ENc value is to 20, the stronger the codon preference. The average ENc value of the *CaD27* gene family is 52.46. The codon adaptation index (CAI) is another important parameter for measuring codon preference, and the average CAI value of the *CaD27* gene family is 0.20 (Figure 4 and Appendix A). According to the results in Figure 2A, the codon preference of the *CaD27* gene family is relatively weak, indicating that during evolution, the *CaD27* gene family may be influenced by genetic drift, resulting from relatively weak selection pressure.

### 2.5. Analysis of Gene Structure and Conserved Motifs in the CaD27 Gene Family

The members of the *CaD27* gene family all contain seven exons (Figure 5). An analysis of conserved motifs reveals a similar number of conserved motifs among most members of the *CaD27* gene family. *CaD27.1* has seven conserved motifs, with motif 5 missing. *CaD27.2* has six conserved motifs, missing motif 6 and motif 8. *CaD27.3* has seven conserved motifs, with motif 3 missing. Further analysis of the conserved protein sequences of CaD27 identifies a domain named DUF4033.

### 2.6. Analysis of Cis-Regulatory Elements in the CaD27 Gene Family Promoter

TBtools was used to extract the gene sequence upstream of 2000 bp as the promoter sequence for the *CaD27* gene family. The results showed that the *cis*-regulatory elements of the three members of *CaD27* can be divided into five categories: phytohormone-responsive elements, light-responsive elements, tissue specificity-related elements, abiotic stress-responsive elements, and transcription factor recognition and binding sites (Figure 6). Among these, the proportion of light-responsive elements is the highest, followed by phytohormone-responsive elements (Appendix A).

The promoter region of *CaD27.3* contains an auxin-responsive element (TGA-box) and an abscisic acid-responsive element (ABRE). The promoter regions of *CaD27.1* and *CaD27.3* include one and three gibberellin-responsive elements (GARE-motif, P-box, and F-box), respectively. The promoter regions of all *CaD27* members contain a salicylic acid-responsive element (TCA-box) and an ethylene-responsive element (ERE). This indicates that *CaD27.3* plays an important role in multiple hormone responses.

Among the light-responsive elements, the proportion of the conserved DNA motif Box 4 is the highest in *CaD27* members. Among the stress responsive elements, the proportion of the osmotic stress response element STRE is highest in *CaD27* members. *CaD27.1* contains three STRE elements, suggesting a role in the induced response of plants to osmotic stress. In summary, the *CaD27* gene family not only plays a normal role in strigolactones’ biosynthesis but may also participate in plant light response, hormone response, stress response, and growth and development activities.

### 2.7. Analysis of Tissue-Specific Expression of the CaD27 Gene Family

The relative expression patterns of the *CaD27* gene family were analyzed using qRT-PCR. The total RNA was extracted from different tissues of the pepper samples, including roots, stems, leaves, flowers, and fruits, and then reverse-transcribed into cDNA. Subsequently, the expression levels of the *CaD27* gene family were quantified by RT-qPCR, using the expression levels in flowers as a control. The results showed differential expression patterns of the three family members in different tissues of pepper (Figure 7). *CaD27.1* exhibited the highest expression level in leaves, while *CaD27.2* was primarily expressed in roots, indicating the significant role of the *CaD27* gene in the growth and development of pepper leaves and roots.

### 2.8. Prediction and Validation of the Interacting Proteins of the CaD27 Gene Family

Referring to the model plant *Arabidopsis thaliana*, a total of 22 potential interacting members with CaD27 gene family proteins were predicted. Among them, CaD27.1 was predicted to interact with 10 proteins, CaD27.2 with 10 proteins, and CaD27.3 with 5 proteins (Figure 8A). However, only CaD27.2 showed high-confidence interactions (>0.900) with nine proteins: CYP711A1, LCY1, RL2, D14, CCD8, CYP97A3, BETA-OHASE_2, BETA-OHASE_1, and CCD7. The high-confidence interacting proteins were validated using yeast two-hybrid experiments. The cloning of genes was performed using a mixture of cDNA from various parts of pepper, such as roots, stems, leaves, flowers, and fruits. CaD27.2 and its interaction partners were successfully cloned.

Full-length coding sequences of 10 genes were cloned into the yeast pGBKT7 vector to assess autoactivation. The AH109 strain was transformed with recombinant plasmids containing the target gene in pGBKT7 and empty pGADT7 vectors. Results indicated that strains carrying pGADT7 empty vectors or those with pGBKT7 fused with Ca27.2, CaCCD7, CaCCD8, CaCYP97A3, CaD14, and CaLCY grew normally on selective DDO medium (SD/-Trp-Leu), but failed to grow on stringent selective media QDO (SD/-Trp-Leu-Ade-His) and QDO/X-α-Gal (SD/-Trp-Leu-His-Ade+X-α-Gal), suggesting no autoactivation (Figure 8B).

Moreover, co-transformed AH109 strains with combinations of pGADT7-CaCCD7+pGBKT7-Ca27.2, pGADT7-CaCCD8+pGBKT7-Ca27-2, pGADT7-CaCYP97A3+pGBKT7-Ca27.2, pGADT7-CaD14+pGBKT7-Ca27.2, and pGADT7-CaLCY+pGBKT7-Ca27.2 could grow normally on a DDO medium, indicating successful co-transformation. Only pGADT7-CaCCD7+pGBKT7-Ca27.2 exhibited growth on a QDO medium and turned blue on a QDO/X-α-Gal medium (Figure 8C). This suggests a specific interaction between CaCCD7 and Ca27.2. In vitro Y2H assays confirmed heterodimer formation between CaCCD7 and Ca27-2 proteins.

## 3. Discussion

Pepper (*Capsicum annuum* L.) is a plant belonging to the Capsicum genus of the Solanaceae family, originating from the Americas and now widely cultivated worldwide. It is an important vegetable and spice crop valued for its compounds such as capsaicin, vitamins, and various minerals, which contribute to its unique spicy taste and aroma [35]. Strigolactones play a significant role in the growth, development, and yield of pepper [5,10], and promote root development, control branching, facilitate symbiosis with mycorrhizal fungi, enhance stress resistance, and influence interactions between plant hormones in the plant. In this study, three members of the *D27* gene family in pepper were identified and named *CaD27.1*, *CaD27.2*, and *CaD27.3*. These genes exhibit differences in their isoelectric points. Bioinformatics analysis revealed differentiation in the amino acid sequence, gene structure, and conserved motifs of the *CaD27* genes during evolution, indicating their distinct functions in plant growth and development. This suggests that the *CaD27* gene family contains different types of genes with varied functions or structures in the specified gene regions. Additionally, it reflects potential transcriptional regulation, gene rearrangement, or other forms of gene variation during the evolutionary process. In summary, the members of the *CaD27* gene family possess a degree of diversity and non-repetition, providing clues for understanding the evolution and diversity of the *CaD27* gene family. Similar genes have been identified in *Oryza sativa* (*OsD27*) and *Arabidopsis thaliana* (*AtD27*) [36,37], which encode β-carotene isomerase, catalyzing the conversion of all-trans-β-carotene to 9-cis-β-carotene [38]. Moreover, the functional role of the *D27* gene has been confirmed in other plants such as *Dendranthema grandiflorum* [39], *Glycine max* [40], and *Medicago truncatula* [41]. In *Oryza sativa*, *D27* encodes an iron-containing protein located in chloroplasts, and is essential for strigolactones’ biosynthesis, capable of suppressing tiller number and increasing plant height [36]. Under phosphorus and sulfur deficiency, *OsD27* is highly expressed, while nitrogen deficiency has no significant effect [42,43,44]. Similarly, *Arabidopsis thaliana* exhibits high expression of *DWARF 27* under phosphorus deficiency, and it is also induced by auxin and abscisic acid [38].

Whether in *Oryza sativa*, *Arabidopsis thaliana*, or *Pisum sativum*, strigolactones can inhibit tillering [3,16]. Phylogenetic analysis of the evolutionary tree shows that pepper’s *D27* has a closer relationship with the *D27* of solanaceous plants and a more distant relationship with *Sesamum indicum*, *Striga asiatica*, and *Arabidopsis thaliana*. It is speculated that the role of *CaD27* in regulating strigolactone synthesis in pepper plants is different from that in *Oryza sativa* and *Arabidopsis thaliana*, mainly by regulating branching patterns and leaf development to structure the aboveground parts of pepper. In addition, the three *CaD27* in pepper are not in the same group and are not collinear, indicating significant differentiation during evolution. By analyzing the gene promoter regions, the potential *cis*-acting elements were identified. The promoter region of the *CaD27* gene contains numerous hormone-responsive elements and light-responsive elements. Differences in regulatory elements in the promoter regions of *CaD27.2* highly expressed in roots, and *CaD27.1*, highly expressed in leaves, may explain their tissue-specific expression. *CaD27.1* has a higher number of regulatory elements related to tissue specificity and phytohormone responsiveness, while *CaD27.2* has more light-responsive elements. The promoter region of the *CaD27* gene contains multiple gibberellin response elements (GARE-motif, F-box, and P-box), ethylene response elements (ERE), and salicylic acid response elements (TCA). Especially, *CaD27.3* contains 10 GARE-motifs, suggesting that the *CaD27* gene family may involve hormone crosstalk between strigolactone biosynthesis regulation and the gibberellin, ethylene, and salicylic acid signaling pathways [45]. The molecular mechanisms of strigolactone and gibberellin signal transduction are quite similar, both relying on protein degradation mediated by α/β hydrolase and E3 ubiquitin ligase [13,34]. Research also shows that the biosynthetic pathway of strigolactone is regulated by gibberellin, with GA1, GA3, and GA4 inhibiting *D27* synthesis and transcription in rice roots, thereby reducing strigolactone analog levels [46]. Simultaneously, strigolactone acts as a regulatory signal for gibberellin, inhibiting abscisic acid accumulation under heat stress and promoting gibberellin synthesis [25]. These results indicate that the interaction between strigolactone and other hormones plays an important role in promoting the growth and development of pepper, improving fruit yield and quality, and enhancing stress resistance.

*CaD27.2* exhibits higher expression levels in the roots and leaves, similar to *GmD27* (*Glycine max*) [40]. This suggests its possible correlation with the regulation of plant structures, such as branching patterns, leaf development, and root establishment. On the other hand, *CaD27.2* shows the highest expression levels in the roots. SLs and their analogs can regulate root development in *Arabidopsis thaliana* [47] and inhibit adventitious root formation in peas [48], indicating its significant role in regulating root establishment, root architecture, and the interaction of plants with soil nutrients (especially phosphorus) and mycorrhizal fungi. The high expression pattern in the roots enables *D27* to effectively participate in the regulation of these biological processes, particularly in response to soil nutrient deficiency, by controlling plant growth strategies through the biosynthesis of strigolactones. Strigolactones play an important role in root growth, shoot branching, and promoting nodule formation [49,50]. The high expression of *CaD27* in both roots and leaves suggests the conservative nature of the regulatory mechanisms of strigolactones in pepper plants.

Research has shown that D27, CCD7, and CCD8 collectively participate in the conversion of cis-β-carotenes into the intermediate compound, cis-cis-xanthoxin. Subsequently, cis-xanthoxin undergoes oxidation by cytochrome P450 monooxygenase encoded by MAX1 and its homologs to generate carlactonic acid. Then, methyltransferase catalyzes the production of methyl carlactonic acid. Finally, lateral branching oxidoreductase catalyzes the synthesis of strigolactone-like compounds [12,37,51]. The β-carotene isomerase regulated by D27 is located upstream of the strigolactone biosynthetic pathway and is involved in the continuous process of strigolactone biosynthesis along with CCD7, CCD8, and others [37]. CCD7 participates in the metabolism of carotenoids by cleaving various 9-cis-carotenoids [15] and also regulates the gibberellin synthesis pathway. The CCD7 protein is involved in the ubiquitination and degradation of the D27 protein. When interaction occurs between D27 and CCD7, the D27 protein is ubiquitinated and degraded, leading to the release of the key enzyme (GA20ox) in the gibberellin biosynthetic pathway, thereby promoting gibberellin synthesis and potentially further impacting strigolactone biosynthesis regulation [13,52,53]. This study has identified the interaction between the D27 protein and CCD7 protein in strigolactone biosynthesis in peppers, which is significance for understanding the mechanism of plant growth regulation.

## 4. Materials and Methods

### 4.1. Experimental Materials

Plant samples were obtained from the Zunyi cultivation base of the Pepper Research Institute, Guizhou Academy of Agricultural Sciences. The pepper variety used was “ZunLa”. The samples were collected from different tissues of the plant during the same period, including the roots, stems, leaves, flowers, and fruits, in November 2023. After collection, the samples were immediately frozen in liquid nitrogen and stored at a standard temperature of −80 °C in the laboratory. Each treatment was performed with three biological replicates.

### 4.2. Identification and Analysis of the CaD27 Gene Family

The whole-genome sequence, proteome data, and genome annotation file of “Zunla” pepper were downloaded from the Pepper Genome Database (http://peppergenome.snu.ac.kr/ (accessed on 25 July 2023)). The hidden Markov model (PF13225.10) of *D27* was obtained from the Pfam database (http://pfam.Xfam.org/ (accessed on 27 July 2023)) to screen pepper protein sequences. The homologous genes in *Capsicum annuum* were identified by performing a BLAST search against the *D27* sequence in *Arabidopsis thaliana* using TBtools (https://github.com/CJ-Chen/TBtools (accessed on 28 October 2023)). Redundant sequences from both search methods were merged and manually curated to remove members with incomplete N/C terminals using NCBI CD-Search (https://www.ncbi.nlm.nih.gov/cdd/ (accessed on 18 November 2023)), resulting in the final identification of the *D27* gene family members in pepper (Appendix A). The chromosome locations of the *D27* gene family members were determined from the pepper genome GFF file using TBtools and named based on their positional order on the chromosome, with visualization facilitated by TBtools [54].

Using Expasy (https://web.expasy.org/protparam/ (accessed on 20 November 2023)), predictions were made for the isoelectric point and relative molecular weight of *D27* members to ascertain their protein physicochemical properties. Transmembrane structure, signal peptides, and subcellular localization were analyzed using TMHMM Server (https://dtu.biolib.com/DeepTMHMM (accessed on 20 November 2023)), PrediSi (https://predisi.de (accessed on 20 November 2023)), and WoLF PSORT (https://www.genscript.com/wolf-psort.html (accessed on 20 November 2023)), respectively. The 2000 bp upstream promoter regions of *D27* were analyzed using plant CARE (http://bioinformatics.psb.ugent.be/webtools/plantcare/html/ (accessed on 20 November 2023)) to predict *cis*-acting elements. MEME (http://meme-suite.org (accessed on 20 November 2023)) was employed with default parameters to analyze the conservative sequences, important functional sites, and motifs of the initially identified full-length CaD27 protein sequence. Tbtools (1.120) software was used to visualize the gene structure of *CaD27*. Codon (W 1.4.4) and EMBOSS (1.4.2) software were utilized to compute the main parameters of codons and the relative synonymous codon usage of the *CaD27* gene family members.

### 4.3. Construction of the CaD27 Gene Family Phylogenetic Tree

The amino acid sequences of *D27* from *Arabidopsis thaliana*, *Nicotiana tabacum*, *Solanum lycopersicum*, *Solanum tuberosum*, ‘CM334’ pepper, ‘Ca59’ pepper, and others were obtained (Appendix A). Multiple sequence alignment of the protein amino acid sequences of *D27* was performed using the neighbor-joining method in MEGA11 (11.0.13) software. A systematic phylogenetic tree was constructed with the bootstrap value set to 1000 and default parameters values [55].

### 4.4. CaD27 Gene Family Collinearity Analysis

The TBtools software’s Advanced Circos function was utilized to conduct collinearity analysis on the *D27* gene family within the ‘Zunla’ pepper genome. Simultaneously, collinearity analysis was performed between D27 members in ‘Zunla’ pepper and those in ‘CM334’ pepper, ‘Ca59’ pepper, *Arabidopsis thaliana*, *Nicotiana tabacum*, *Solanum lycopersicum*, and *Solanum tuberosum*.

### 4.5. CaD27 Gene Expression Pattern Analysis

Total RNA was extracted from various tissues of ‘Zunla’ using the RNA extraction kit (TIANGEN BIOTECH Co., Ltd., Beijing, China). Approximately 50–100 mg of RNA was obtained per sample, following the manufacturer’s instructions, with each sample processed in triplicate. The RNA was then reverse-transcribed into first-strand cDNA using the Trans Script^®^ II All-in-One First-Strand cDNA Synthesis Super Mix for qPCR (TransGen Biotech Co., Ltd., Beijing, China). Each cDNA sample was diluted to 35 µL with nuclease-free water and stored at −20 °C for subsequent use in qRT-PCR. Specific primers were designed for the *CaD27* gene sequence to examine its expression patterns, with Actin (XM_016688188.2) serving as the internal reference gene (Appendix A). The qPCR reaction included an initial denaturation step at 95 °C for 30 s, followed by 40 cycles of denaturation at 95 °C for 5 s and annealing/extension at 60 °C for 30 s. Experiments were conducted using a German Jena fluorescent quantitative PCR instrument, with each biological sample assayed in triplicate. Relative expression levels were calculated using the 2−ΔΔCt method [56], and graphs were generated using TBtools (1.120).

### 4.6. Prediction and Validation of Interacting Proteins of the CaD27 Protein

The amino acid sequences of *CaD27* gene family members were submitted to the STRING website (https://string-db.org/ (accessed on 20 December 2023)) to predict protein-protein interactions, using *Arabidopsis thaliana* as the reference genome with a medium confidence threshold (Required score >0.400) (Appendix A). Predicted proteins from *Arabidopsis thaliana* were aligned to the pepper genome to identify the potential interacting proteins of the *CaD27* gene family in pepper. Candidates showing high interaction confidence (>0.900) were selected for yeast two-hybrid experiments to validate these interactions.

Using the mixed cDNA of the ‘Zunla’ whole tissue as a template, *CaD27.2* and the coding sequences (CDS) of the interacting proteins (Appendix A) were cloned. The amplified target gene was constructed on pGADT7 and pGBKT7 linearization vectors by the In-Fusion method. After being transformed into competent *E. coli* cells, positive clones were screened and sequenced by Sangon Biotech. Plasmids with confirmed sequences were extracted using the Mini Plasmid Kit (Tiangen, Beijing, China) for subsequent yeast two-hybrid experiments [56].

Transform constructs, including pGADT7+pGBKT7-*Ca27.2*, pGADT7-*CaCCD7*+pGBKT7, pGADT7-*CaCCD8*+pGBKT7, pGADT7-*CaCYP97A3*+pGBKT7, pGADT7-*CaD14*+pGBKT7, and pGADT7-*CaLCY*+pGBKT7, were introduced into the yeast host strain AH109, plated on a DDO (SD/-Trp-Leu) medium, and incubated inverted at 30 °C for 3-5 days. Colonies were subsequently picked for colony PCR verification and spot on QDO (SD/-Trp-Leu-Ade-His) and QDO/X-α-Gal (SD/-Trp-Leu-His-Ade+X-α-Gal) media, and incubated inverted at 30 °C to detect autoactivation.

Furthermore, constructs including pGADT7-*CaCCD7*+pGBKT7-*CaD27.2*, pGADT7-*CaCCD8*+pGBKT7-*CaD27.2*, pGADT7-*CaCYP97A3*+pGBKT7-*CaD27.2*, pGADT7-*CaD14*+pGBKT7-*CaD27.2*, and pGADT7-*CaLCY*+pGBKT7-*CaD27.2* were transformed into yeast strain AH109, plated on a DDO (SD/-Trp-Leu) medium, incubated inverted at 30 °C for 3–5 days, followed by colony PCR verification and spotting on QDO (SD/-Trp-Leu-Ade-His) and QDO/X-α-Gal (SD/-Trp-Leu-His-Ade+X-α-Gal) media, and incubated inverted at 30 °C to assess interactions.

## 5. Conclusions

The *D27* gene family is hypothesized to play pivotal roles in plants. In addition to its involvement in the secondary metabolism related to strigolactone biosynthesis, *CaD27* may also influence plant photomorphogenesis, hormone responses, stress tolerance, and overall growth and development of pepper leaves and roots. Moreover, these three members of the *CaD27* gene family exhibit notable diversity and variation throughout evolution. *CaD27* significantly contributes to the growth and development of pepper leaves and roots. Further investigation into the functions and regulatory mechanisms of the *D27* gene family is crucial for elucidating the molecular mechanisms underlying strigolactone’s role in plant growth and development.

## Figures and Tables

**Figure 1 plants-13-02070-f001:**
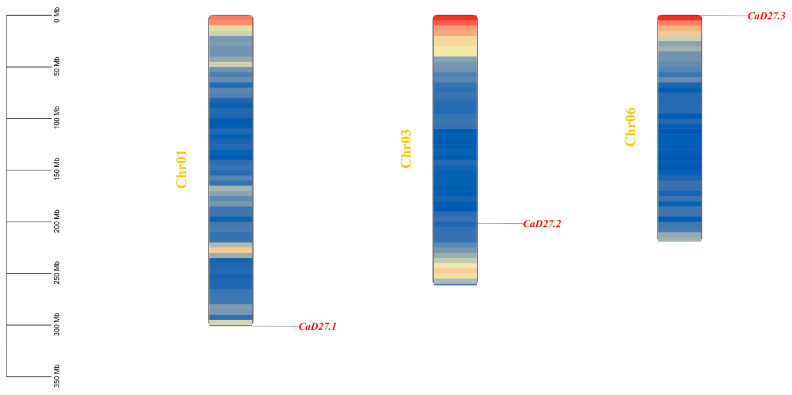
The distribution of 3 *CaD27* genes on the chromosomes. The scale on the left is used to estimate the chromosome lengths. Blue to red on the chromosome indicates gene density.

**Figure 2 plants-13-02070-f002:**
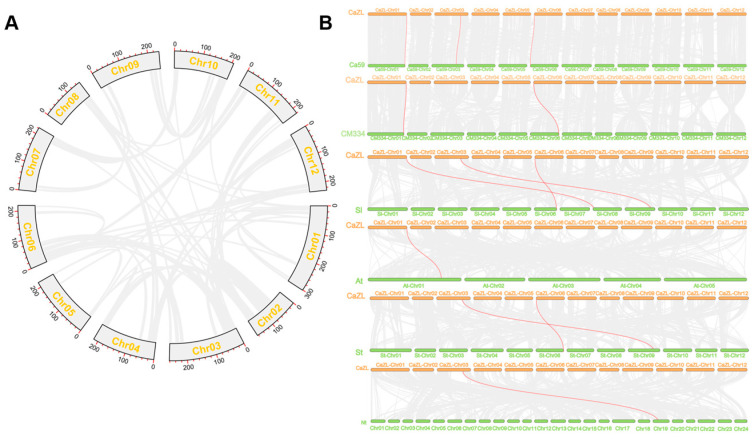
Collinearity analysis of the pepper (*Capsicum annuum* L.) *D27* Gene Family. (**A**) Chromosomal distribution of all gene pairs in pepper, but *CaD27* has no collinear gene pairs within the genome. (**B**) Collinearity analysis of *CaD27* between ‘Zunla’ pepper (CaZL) and ‘Ca59’ pepper (Ca59), ‘CM334’ pepper (CM334), *Solanum lycopersicum* (Sl), *Arabidopsis thaliana* (At), *Solanum tuberosum* (St), and *Nicotiana tabacum* (Nt), respectively. Note: Gray lines represent all gene pairs based on a collinearity analysis of GFF files, while red lines represent *D27* gene pairs with collinearity.

**Figure 3 plants-13-02070-f003:**
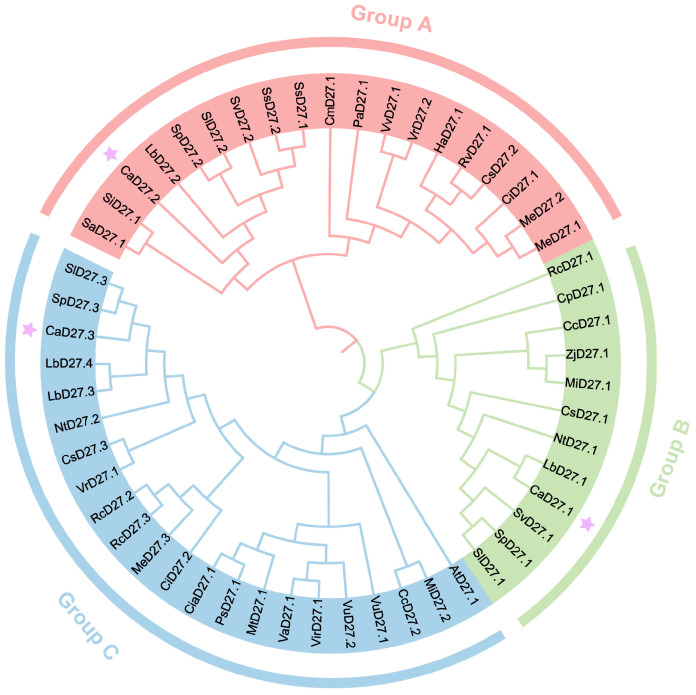
Phylogenetic tree of the D27 family proteins in *Capsicum annuum* and other plant species, where purple asterisks are used to highlight the position of the CaD27 gene family.

**Figure 4 plants-13-02070-f004:**
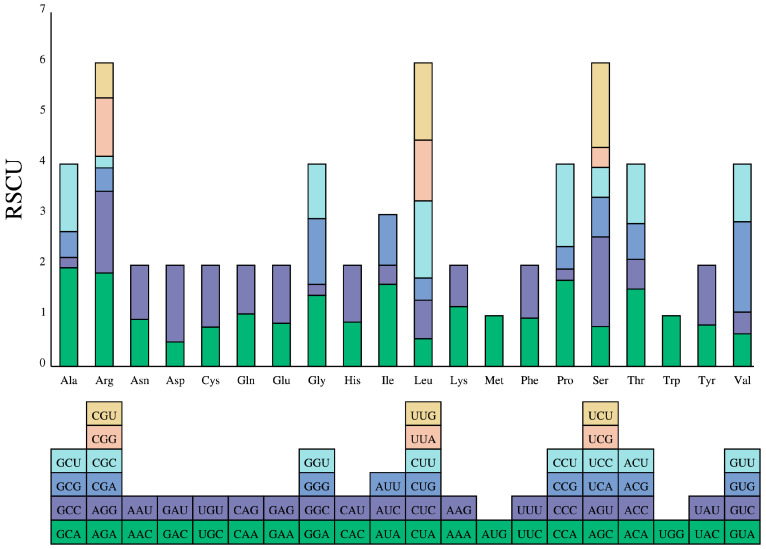
Codon preference of *CaD27* gene family in “Zunla” pepper. The horizontal axis codon represents the codon type used for amino acids in the same column, and the color of the codon below the horizontal axis corresponds to the RSCU (relative synonymous codon usage) value of the same color above the horizontal axis.

**Figure 5 plants-13-02070-f005:**
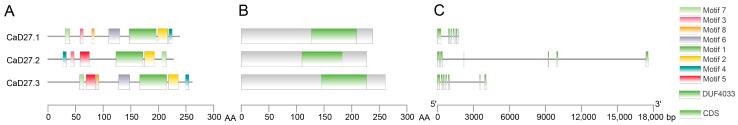
Motifs, protein domain, and gene structure of the *CaD27* gene family. (**A**) Conserved motifs of the CaD27 gene family. (**B**) The protein domain of the CaD27 gene family. (**C**) The gene structure of the *CaD27* gene family. Note: the coordinates of motifs and protein domain were in amino acid residues (AA), and gene structure coordinates were in base pairs (bp).

**Figure 6 plants-13-02070-f006:**
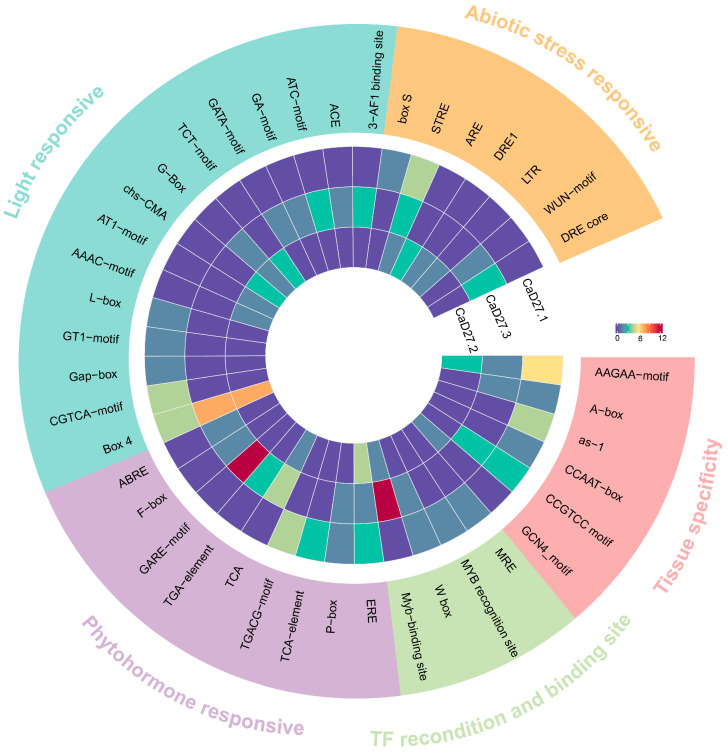
Analysis of *cis*-acting elements in the promoter region of the *CaD27* gene family. These *cis*-acting elements were classified according to their functions and were distinguished by different colors, with their respective categories marked on the outside of the group. The heatmap data were centralized for more intuitive comparison.

**Figure 7 plants-13-02070-f007:**
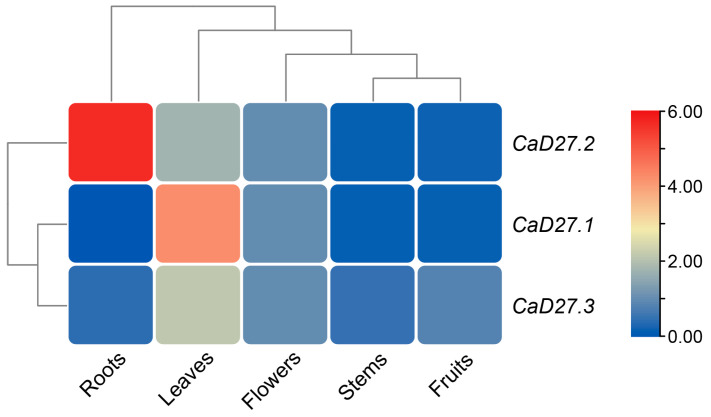
The relative expression patterns of the *CaD27* gene family in different tissues during the same period. Using *CaD27* expression in flowers as a control, the fold difference in expression in other tissues was compared to flowers. Sampling was collected at fruit maturity, included mature fruit, flowers at anthesis, fully expanded leaves, and healthy stems and leaves.

**Figure 8 plants-13-02070-f008:**
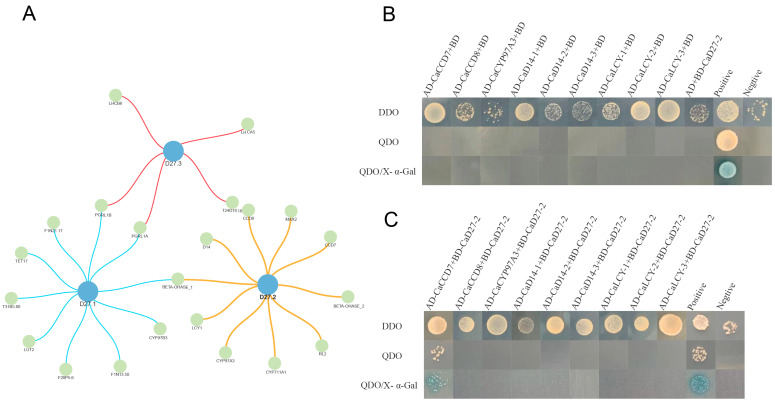
Analysis of interacting proteins in the CaD27 gene family of pepper (*Capsicum annuum* L.). (**A**) Prediction of the interacting proteins of the CaD27 gene family based on STRING Database. (**B**) Preliminary screening of CaD27.2-interacting proteins in pepper based on yeast autoactivation. (**C**) Validation of CaD27.2-interacting proteins in pepper based on yeast two-hybrid assay.

## Data Availability

The original contributions presented in the study are included in the article/Appendix A, further inquiries can be directed to the corresponding authors.

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
