# Peer review of "Genome-Wide Identification and Expression Characterization of the D27 Gene Family of Capsicum annuum L."

_plants, 2024, doi:10.3390/plants13152070_

Round 1
Reviewer 1 Report
Comments and Suggestions for Authors
In this study “Genome-wide identification and expression characterization of the D27 gene family of Capsicum annuum L.” the authors identified and analyzed three CaD27 genes in pepper, revealing their physicochemical properties, phylogenetic relationships, gene structures, promoters, and tissue-specific expression patterns.
In this manuscript, the Introduction is well written, the results are clearly explained, and the Discussion is also well written. However, there are some issues I would like the authors to address especially in the Materials and methods section.
1. Revise the Material and methods section, ensuring that the methods are described in terms of the actual procedures followed in the study, rather than as a protocol.
2. The use of only Actin as an internal control for gene normalization raises concerns, as relying on a single reference gene for quantitative PCR (qPCR) is typically not recommended.
3. In the text and figures put all Latin names of plant species, e.g. Capsicum annuum, Arabidopsis thaliana, in italics.
4. In Figure 4, italicize "CaD27" gene to differentiate it from proteins.
5. I recommend labeling Figure 5 with alphabets to enhance the clarity and understanding of the figures.
6. Italicize DWARF 27 in line 256 if you are writing about gene.
Comments on the Quality of English LanguageEnglish requires minor revision.
Reviewer 2 Report
Comments and Suggestions for Authors
In my opinion, the study titled "Genome-wide identification and expression characterization of the D27 gene family of Capsicum annuum L." by Wu et al. is highly relevant to its research area, especially regarding the investigation of the D27 gene family that plays a crucial role in plants. However, the Materials and Methods section needs improvement, particularly in the experimental materials. The number of plants used is unclear, and the field experiment details are also vague. The captions for Figures 1, 3, and 7 need to be improved. However, I have a few suggestions for this manuscript:
-
Lines 10-24: The abstract must be improved as it lacks context and does not adequately explain the importance of Capsicum annuum in this research.
-
Line 32: The statement "the formation of plant architecture is controlled by various factors" requires more citations for each example provided.
-
Line 59: A citation is missing after “stress response.”
-
Line 77: Add "Capsicum annuum L." after "Pepper."
-
Lines 102-121: In my opinion, this paragraph seems more appropriate for the discussion section rather than the results.
-
Line 144: "Capsicum annuum" should be italicized (please check the entire manuscript for this).
-
Line 200: How many plants were used for RNA extraction? This information is not specified in the methods section.
-
Line 211: "Arabidopsis thaliana" should be italicized (same comment as above).
-
Lines 319-325: This paragraph needs improvement. It lacks information: How many plants were analyzed? Why is the month of sampling highlighted? Is it important to know? Why? The extraction from different tissues likely required different extraction protocols, correct?
-
Line 338: There is a new version of the TBtools software (Chen et al., 2023), so the citation should be updated.
-
Lines 394-397 and 400-404: Has this protocol been published before? If so, cite the article.
Reviewer 3 Report
Comments and Suggestions for Authors
JP – Review.
Manuscript ID: plants-3047658
“Genome-wide identification and expression characterization of the D27 gene family of Capsicum annuum L.”
The authors of the manuscript "Genome-wide identification and expression characterization of the D27 gene family of Capsicum annuum L" analyzed the D27 genes involved in the synthesis of strigolactones in Capsicum annum. The analyzes included the identification of homologues of this gene in the C. annuum genome, their chromosomal location, the analysis of collinearity of these genes in the genome of one of the C. annum cultivars and the analysis of collinearity of these genes between different C. annuum cultivars and between C. annum and several selected plants. Various bioinformatics tools were used to comprehensively analyze the genes, proteins, and promoter regions of C. annum D27 genes. In addition to bioinformatic analyses, the expression of C. annuum D27 genes was examined using the RT-qPCR method and the yeast two hybrid system was used to identify proteins interacting with one of D27 proteins from C. annuum.
Much work has been done and the results should be of interest to Plants readers, but the presentation of the results needs improvement. Currently, the reader often has to reconstruct the results on his own, sometimes using data from three sources: figures, text and additional tables. The Authors treat the figures as illustrations of their theses, not as reports of results; many figures have incomplete descriptions or even lack captions and explanations. The results have to be presented in a precise and understandable way for the reader. Below is a list of my most important remarks:
1 - lack of data from the results, lack of description or imprecise description of the figures presenting the results. I point out the most important of these shortcomings below, but this applies to all Result.
Chapter 2.1 - there is no accession number of protein used to search for C. annuum D27 homologs, there is no data on E value or % of identity of the positive hits.
Chapter 2.2 – there is no description to Fig. 2A, the authors should describe what the visible elements, e.g. gray lines, mean. Since the figure is to show that the analyzed genome lacks collinear regions with the CaD27 genes, these genes should be indicated. The statement on line 108 seems unjustified - how does the lack of collinearity indicate different functions of genes from the family and different structures of these genes and what does "in the determined regions" mean? I think the authors should explain it. The abbreviation D27 (italics) is used both to mean the D27 gene and the the D27 gene family (it concerns also other chapters), which is often confusing. For example in the sentence of line 116 it is not clear what the sense of the sentence is. Similarly, in the case of line 120, it is not clear which of the D27 genes we are talking about, and what "pepper" means here, is it one of the varieties, and which one? This imprecise use of terms occurs throughout the all manuscript and forces the reader to guess the meaning of the authors' arguments or thesis. Caption under Fig. 2B is incorrect - the pepper varieties mentioned are not different species, but different varieties.
Chapter 2.3 - Common plant names are used throughout the chapter, and the text directs the reader to Table S3 for explanation of the identifiers of the analyzed proteins. However, Table S3 contains the Latin names of the plants on which the protein identifiers, used in Fig.3, are also based. As a result, the reader cannot connect the authors' statements with the results of the phylogenetic analysis, i.e. with the phylogenetic tree presented in Fig. 3.
Chapter 2.4 – the results are presented mainly on Fig. 4, but there is no explanation to this figure. Please explain what RSCU means on the vertical axis, please also describe the horizontal axis, codons in the boxes below the graph, and the meanings of the colors. Since the article discusses 3 varieties of peppers, please specify which of them means "pepper" from the caption.
Chapter 2.5 – this chapter needs improvement as it is imprecise, it seems to concern both D27 genes and D27 proteins, but they are not distinguished. The reader must guess for himself whether a given sentence refers to a gene or a protein. Also it is not known whether the term "motif" refers to a gene, a coding sequence, or a protein. Fig. 5 is also not described properly, it is not known which of the diagrams refers to genes, cds, and which to proteins. All three horizontal axes have no units, but they all have their ends labeled 3' and 5', suggesting that they all represent nucleotide sequences - genes or cDNA. Moreover, the Authors cannot state that the "gene family has 6-7 conservative motifs” (lane 165) because it either has 6 or 7 of them.
Chapter 2.6 - The authors do not provide a list of the locations of potential regulatory elements, or even their number, or information on the criteria for the similarity of the identified regulatory elements to sequences from the database. The reader only has access to Fig. 6, which does not even contain a description of the legend. Therefore, the reader cannot estimate how many specific regulatory elements have been identified in the promoter of a given gene. The authors indicate and discuss the most important categories of regulatory elements based on their number, but they do not state whether this concerns the number of different regulatory elements in this category or the number of places with such elements in the promoter of given gene. Again, the abbreviation D27 (italics) is used both to mean the D27 gene and the D27 gene family, which is often confusing, for example in the sentences of lines 173, 191, 192, etc.
Chapter 2.7 - based on the description in Chapter 4.5, it appears that relative expression results are presented. However, this is not described in this chapter, neither in the description of Fig. 7. The accession nr. of the sequence that was used to design primers for the reference gene is not given neither here nor in the chapter 4.5. The description of Fig. 7 is insufficient - no description of the scale. There is no description for calculating average values ​​and SD for relative expression, and the reader does not have access to these data, so it is difficult to assess the accuracy of values from Fig. 7. Throughout the chapter, the authors describe leaves, roots, etc. as tissues. It is also worth noting that due to the inaccurate description of the plant material, it is difficult to comment on the authors' claims that the CaD27 genes are involved in the development of leaves and roots (lines 206-206), because in leaves that have completed the expansion phase, developmental processes are not very active.
Chapter 2.8 – this entire paragraph needs improvement. Abbreviations and identifiers used here should be clarified and explain. Currently they are not provided in this chapter, M&M or in the supplementary materials. This also applies to media names, I believe the reader should be informed which ones are controls and what type. Moreover, the media have different abbreviations in the text and in Fig. 8 (QDO/X vs. QDO + alpha X Gal). Tthere are also differences in names for the yeast lines and constructs between the text and on the figure: pGADT7-DaCCD7+pGBKT7-Da27-2 (in the text) vs AD-CaCCD7+BD-Ca27-2 (on Fig. 8C), and so on.
The statement that CaD27 proteins interact with 25 proteins (line 211) is incorrect. Some of these proteins interact with two CaD27 proteins, hence the sum of these proteins is 22 and not 25, see Fig. 8A.
In my opinion, the Authors should better explain to the reader why the growth of the pGADT7-DaCCD7+pGBKT7-Da27-2 yeast line on QDO and QDO+X-alpha-Gal media indicates the interaction of CaD27.2 proteins and DaCCD7.
Fig. 8 - Fig. 8C is not mentioned in the caption to Fig. 8. The description of Fig. 8 is insufficient and imprecise. For example, Fig. 8A shows prediction results, not predictions. I think it should also be mentioned that proteins interacting with CaD27 proteins are color-coded and explain what the colors mean. Description to Fig. 8B is incorrect! This figure does not present validation of the interactions between the CaD27-2 protein and the proteins that are presumed to interact with it. Moreover Fig. 8B and 8C concern exclusively one protein, CaD27-2 and not the whole protein family.
2 - Authors' thesis or conclusions are presented together with the results, in the section Results; moreover, these conclusions are often not supported by the authors' results.
Chapter 2.2 - I cannot agree with the statements from lines 108 and 120, I think that the Authors should arguing them better. And preferably this should be done in the Discussion. Regarding the statement from line 120, which of the three CaD27 proteins does it refer to and what proves greater conservation between the CaD27 proteins and the tomato homologs compared to the conservation between the CaD27 proteins and the potato homologs? In my opinion, the results presented on Fig. 2B suggest that in plants from the same family as CaZL, i.e. from the Solanaceae family, each of these three D27 genes can be eliminated from the genome, but at least one of them must be present in the genome.
Chapter 2.3 – The authors present here not only the results, but also their theses and conclusions. Moreover, these conclusions are often presented without argument. For example: on lines 135-137, the authors write that the CaD27.2 protein has functional similarity to purslane D27, with no other arguments than the phylogenetic tree, moreover there is no such plant name on the tree from Fig.3! I also cannot agree with the authors' statement regarding the close phylogenetic relationship between the CaD27.2 protein and the homologous sesame protein (lines 135-137). In my opinion, based on Fig. 3, the CaD27.2 protein is most closely related to the D27 homologs of Solanaceae proteins from group A. They forms a sub-clade within group A. The SiD27.1 protein from sesame, Sesamum indicum, does NOT belong to this sub-clade. The SiD27.1 protein from sesame is most closely related to the SaD27.1 protein from Striga asiatica, and together they form the sister group to the sub-clade of D27 Solanaceae proteins from group A proteins. In my opinion, based solely on Fig. 2, all three CaD27 proteins are most closely related to homologs from the Solanaceae family, which is consistent with plant phylogeny.
I would like to draw the authors' attention to the fact that their phylogenetic analysis (Fig. 3) indicates that the CaD27.3 protein from Capsicum annuum and At27.1 from A. thaliana belong to clade C. This means that At27.1 is more closely related to CaD27.3 than to CaD27.1 and CaD27.2. However, Fig. 2B indicates that ATD27 from Chr01, presumably AtD27.1 is more closely related to CaD27.1 than to CaD27.2 or CaD27.3. Can the Authors comment on this?
Chapter 2.4 – based on codon preferences, the authors conclude that the CaD27 genes show "relatively low levels of gene expression." Does this agree with the expression data, where the CaD27.2 gene is 5-6 times more expressed in roots than actin?
Chapter 2.6 – also here a significant part of section 2.6 is a discussion of the results. Some statements from this discussion seems to be illogical. For example, the Authors write that the STRE element is the most frequent element in the group of "stress responsive elements" in the CaD27 gene promoters. And further they say that in the promoter of one of the CaD27 genes there is one element of this type! I suggests to analyze each of the three promoters separately.
3 - The Discussion needs improvement, in many cases the authors quote statements without providing references - see lines 240-243, 261-264.
In the Discussion, the Authors refer to D27 protein or genes from plants, which are not used in their phylogenetic analysis, collinearity study, or in any other analyses. So where do they conclude that they are similar to CaD27 proteins? What is the % similarity between these proteins and CaD27 proteins?
Some of the theses in the Discussion seem unfounded, even inconsistent with the Authors' results. For example - in lines 258-260, the Authors state that CaD27 proteins are closely related to homologues from A. thaliana and legumes (Phabaceae). This is not supported by the Authors' results. In my opinion, the Authors' phyologenetic analysis (Fig. 3) shows that D27 proteins from plants from the Phabaceae family are located far from CaD27 homologs. The same applies to the D27.1 protein from A. thaliana.
Regarding the statements from lines 285-287 - I think that the similarity between the CaD27.1 and OsD27 genes seems rather weak. According to the work of [36], the expression of the OsD27 gene in leaves is similar to its expression in roots and much lower than in culms, panicles and other organs. However, CaD27.1 expression in leaves is much higher than in roots and stem, flowers and fruits. Moreover, the authors do not precisely characterize the plant material used, and the phase of leaf development is unknown. Meanwhile, in mature leaves, after the expansion phase has ended, the processes related to leaf development or lateral branching are not very active. I propose to check whether OsD27 is in the same group as CaD27.1, this may support the presumed similarity of their promoters and, consequently, a role in developmental processes.
4 – The nomenclature needs improvement. For example, the Authors use both scientific and colloquial names of plants. In Fig. 2, identifiers based on Latin names were used, but to explain them, the reader is directed to Table S3. Meanwhile, in the discussion of Fig. 2, the Authors use common plant names such as "sesame" and "purslane", for which no Latin names are provided. As a result, the reader either cannot use this data or has to use four sources: figure, text, table and encyclopedia. Next - the Authors use plant names that are neither scientific nor colloquial - "Arabidospis". Incorrect terms are used such as: "highly homologous" whereas homology is a feature that is not gradable, genes or organs are either homologous or not, "Systematic Evolutionary Analysis" - I suggest to change for "phylogenetic analysis". Then, "leaves" and "roots" are named "tissues" (Chapter 2.7), varieties off pepper are not species (Fig. 2 caption), “in vitro” instead of “in vivo” (lane 230), etc, etc. Moreover Authors very often use terms that mean nothing and do not provide the reader with any information, such as: “certain”, “rather”, “etc, etc”. I would like to point that the Authors should present their results in a precise and understandable way for the reader.
5 - In the introduction, information about D27 proteins, their functions, whether they are enzymes and what enzymatic activity they have is insufficient.
6 - The Materials and Methods section requires deep linguistic improvement. Currently, some sub-chapters require the reader to guess the meaning of the sentences. The description of some procedures is chaotic, often providing irrelevant and too detailed information, but there is no description of their important stages, for example – for genetic constructions, whether they have been verified by sequencing. Sub-chapter 2.6 on the yeast two-hybrid system does not contain any reference to this method, and the description of the procedure is completely incomprehensible.
Additional remark
I believe that authors can increase the attractiveness of their work by comparing their own results. For example, by checking whether the promoter of a gene with high expression in roots differs in the presence of regulatory elements from the promoter of a gene with high expression in leaves. Similarly, it is worth analyzing the differences in codon frequency between genes with high and low expression.
To sum up, in my opinion, the results presented by the Authors are original and should be interesting for Plants readers, but the manuscript requires deep improvement in terms of language, presentation of the results and their discussion.
Sincerely – JP
..............................................
JP – Review. Concern:
Manuscript ID: plants-3047658
“Genome-wide identification and expression characterization of the D27 gene family of Capsicum annuum L.”
Comment on the quality of English
The manuscript contain sections which are is mostly written in correct language whereas the other parts require deep correction. There are also sections where the reader has to guess the sense of the text.
Moreover the following raise objections:
1 - using terms that are inconsistent with scientific terminology, such as: "Arabidopsis", "highly homologous".
2 - using imprecise terms (certain, etc., rather) that do not allow the reader to understand or verify the authors' theses.
JP.
Comments on the Quality of English LanguageJP – Review. Concern:
Manuscript ID: plants-3047658
“Genome-wide identification and expression characterization of the D27 gene family of Capsicum annuum L.”
Comment on the quality of English
The manuscript contain sections which are is mostly written in correct language whereas the other parts require deep correction. There are also sections where the reader has to guess the sense of the text.
Moreover the following raise objections:
1 - using terms that are inconsistent with scientific terminology, such as: "Arabidopsis", "highly homologous".
2 - using imprecise terms (certain, etc., rather) that do not allow the reader to understand or verify the authors' theses.
JP.
